# Current Clinical Applications of PSMA-PET for Prostate Cancer Diagnosis, Staging, and Treatment

**DOI:** 10.3390/cancers16244263

**Published:** 2024-12-21

**Authors:** Franz von Stauffenberg, Cédric Poyet, Stephan Beintner-Skawran, Alexander Maurer, Florian A. Schmid

**Affiliations:** 1Department of Urology, University Hospital Zurich, University of Zurich, 8091 Zurich, Switzerland; franz.vonstauffenberg@usz.ch; 2Department of Urology, Stadtspital Triemli, 8063 Zurich, Switzerland; cedric.poyet@stadtspital.ch; 3Department of Nuclear Medicine, University Hospital Zurich, University of Zurich, 8091 Zurich, Switzerland; stephan.beintner-skawran@usz.ch (S.B.-S.); alexander.maurer@usz.ch (A.M.)

**Keywords:** prostate cancer, PSMA-PET, staging, diagnosis, radical prostatectomy, biochemical recurrence, mCRPC, radioligand therapy, PSMA-RLT, [^177^Lu]Lu-PSMA

## Abstract

Prostate-specific membrane antigen positron emission tomography (PSMA-PET) has emerged as a highly sensitive imaging modality for prostate cancer (PCa), outperforming conventional methods like CT, MRI, and bone scintigraphy. Over the past decade, PSMA-PET has gained widespread clinical adoption, particularly for staging intermediate- and high-risk PCa, detecting biochemical recurrence, and identifying metastatic disease. Its use of targeted radiotracers enables the detection of even small metastases, offering detailed tumor assessments and supporting advanced therapeutic strategies, especially in radioligand therapy for advanced cases. Integrated into leading clinical guidelines, PSMA-PET has significantly impacted diagnostic workflows and treatment planning. This review explores the evolution of PSMA-PET diagnostics and therapy, addressing its benefits, its integration into clinical practice, and its transformative potential. While limitations remain, PSMA-PET represents a critical step forward in the management of prostate cancer.

## 1. Introduction

Conventional imaging methods for staging prostate cancer include computed tomography (CT), magnetic resonance imaging (MRI), and bone scintigraphy (BS). In the last decade, prostate-specific membrane antigen positron emission tomography (PSMA-PET) has emerged as a more sensitive and specific imaging technique compared to conventional techniques. It has shown significant value in the staging of patients with the newly diagnosed PCa of intermediate- and high-risk groups, as well as in the setting of biochemical recurrence (BCR) or metastatic disease. By utilizing radiotracers that bind specifically to PSMA-expressing cells, and by colocalizing them in conjunction with morphologic cross-sectional imaging techniques (CT or MRI), a more precise diagnostic evaluation can be achieved, enabling the identification of even smaller metastases and providing a detailed assessment of the local tumor environment. The most commonly used radiotracers in PSMA-PET imaging are [^68^Ga]Ga-PSMA-11 and [^18^F]-PSMA-1007 [1,2,3]. The enhanced diagnostic capability not only facilitates more accurate diagnostics but also supports advanced therapeutic strategies. It enables metastasis-directed therapy (MDT) in cases of oligometastatic disease and offers systemic treatment options, such as radioligand therapy. Therapy with [^177^Lu]Lu-labeled PSMA ligands for metastatic, castration-resistant PCa represents a breakthrough in treatment and complements the described diagnostic innovations.

This review aims to explore the evolution of PSMA diagnostics and therapy while also comparing the current recommendations found in the leading clinical guidelines.

## 2. Imaging Modalities and PSMA Targeting in PCa

PSMA is a transmembrane protein with increased expression on the surface of PCa cells. Radiolabeled PSMA ligands are small molecules binding specifically to PSMA. They are internalized into cancer cells and therefore, accumulate in the tumor. While PSMA is present in lower levels in normal tissues such as the kidneys, liver, salivary glands, intestines, and ganglia of the autonomic nervous system, its expression is notably higher in PCa cells. Unbound radiotracers are—depending on the specific ligand used—primarily excreted via the kidneys and urinary tract or the liver. When interpreting imaging results, it is important to consider the physiological uptake and excretion pathways. The small size and low molecular weight of PSMA allow for excellent tissue penetration and effective diffusion into solid lesions [4,5,6].

The first significant step in clinical application occurred in 2008 with the introduction of the small-molecule PSMA inhibitors [123I]I-MIP-1972 and [123I]I-MIP-1095. The first PSMA-PET imaging in humans was conducted in 2010 using the ligand [^18^F]-DCFBC. The clinical breakthrough came in 2011 with the establishment of [^68^Ga]Ga-PSMA-11 for PET imaging and [131I]I-MIP-1095 for the radioligand therapy of metastatic PCa. Over time, [^68^Ga]Ga-PSMA-11 and [^18^F]-PSMA-1007 have become globally recognized in diagnostics, while [^177^Lu]Lu-PSMA-617 has replaced [131I]I-based radioligands in radioligand therapy [4,7,8]. While many studies are based on [^68^Ga]Ga-PSMA-11, [^18^F]-labeled PSMA ligands are becoming increasingly important due to their favorable logistics and availability with longer half-life, cyclotron-production and—owing to their lower intrinsic positron energy—enhanced image quality, depending on the clinical context. However, in terms of diagnostic accuracy, prospective and retrospective comparison studies, along with meta-analyses, have not shown any significant advantage of [^18^F]-PSMA over [^68^Ga]Ga-PSMA-11, suggesting that both are considered equally effective [9,10,11,12,13,14]. The determining factor in the choice of tracer currently remains its availability [5,9,15,16,17].

In combining PSMA-PET with cross-sectional imaging techniques, no definitive advantage has been found for either of the methods (PSMA-PET/CT or PSMA-PET/MRI). A systematic review including a meta-analysis by Evangelista et al. included 23 studies involving 2,104 patients and revealed that PSMA-PET/MRI is superior to both multiparametric MRI (mpMRI) and Choline-PET in primary staging. In a comparison of PET/CT vs. PET/MRI, PET/MRI showed a slight advantage in detecting lymph node metastases, likely due to the longer tracer accumulation time during the extended MRI process compared to the shorter CT scan. However, for detecting bone metastases, PET/CT is more efficient than PET/MRI, particularly when PSA levels exceed 2 ng/mL [18]. Consequently, the selection of the appropriate cross-sectional imaging technique should be tailored to each individual case.

For the standardized reporting of disease extent in histologically confirmed PCa, the PROMISE framework (Prostate Cancer Molecular Imaging Standardized Evaluation) was introduced in 2018 and updated in 2023, integrating findings from whole-body PSMA-PET imaging. PROMISE incorporates the molecular imaging TNM (miTNM) staging system, which categorizes tumor, node, and metastasis stages and quantifies disease burden using metrics such as the total tumor volume and Standardized Uptake Value (SUV), a semi-quantitative measure of tracer uptake in the region of interest. The framework is designed to identify clinically relevant disease stages and allows for the objective comparison of PSMA uptake intensity across lesions. Key features include the use of SUVmax and SUVmean for quantifying PSMA expression, supporting therapy planning and response evaluation, and enabling more precise staging and prognostication [19,20]. The PROMISE criteria have been incorporated into the standardized reporting guidelines of PSMA-PET findings by the European Association of Nuclear Medicine (EANM) [21]. Further, SUV enables the evaluation of the aggressiveness of primary PCa, establishing PSMA expression as an indicator of tumor aggressiveness [22,23].

## 3. Assessment of Loco-Regional Findings and Prostate Biopsy Planning

Primarily, PSMA-PET is used for staging following biopsy-proven PCa diagnosis. However, there is an ongoing discussion about the extent to which PSMA-PET can also be utilized for assessing loco-regional findings, such as in biopsy planning, which is typically carried out using mpMRI. In a multicenter cohort of 72 patients, PSMA-PET/CT-guided biopsies (with either [^68^Ga]Ga-PSMA-11 or [^18^F]-PSMA-1007) showed significantly higher detection rates for PCa compared to systematic biopsies and showed slightly better performance than MRI-guided biopsy in detecting PCa in biopsy-naïve patients. In the repeated biopsy setting, PSMA-PET/CT-guided biopsy slightly surpassed systematic biopsy but was inferior to MRI-guided biopsy. An SUVmax cut-off of 4.8 was most effective for detecting clinically significant PCa. The authors conclude that PSMA PET/CT can effectively guide the biopsy procedure in biopsy-naïve patients with clinically suspected disease, particularly in cases with high-risk clinical features and negative or inconclusive MRI findings [24].

Various studies suggest that the combination of PSMA-PET with mpMRI (PSMA-PET/MRI) improves accuracy compared to PSMA-PET/CT or mpMRI alone, although some reports also indicate a reduction in specificity [25,26,27]. A meta-analysis by Chow et al. states that PSMA-PET/MRI can enhance sensitivity in detecting extraprostatic extension (78.7% vs. 52.9%) and seminal vesicle invasion (66.7% vs. 51.0%) compared to mpMRI. In contrast, PSMA-PET/CT is less effective than mpMRI for these particular assessments [28]. However, to date, only one prospective comparison study with validation through whole-mount pathology has been conducted [29]. This study found that combining [^68^Ga]Ga-PSMA-11-PET/CT with mpMRI improved the delineation of tumor extent compared to using PSMA-PET/CT or mpMRI alone. In this study, segment-level analysis refers to examining predefined prostate areas for PCa localization, while lesion-level analysis focuses on PCa detection. A total of 48% of the segments were positive in final pathology while 35%, 33%, and 46% of the segments were detected by PSMA-PET/CT, mpMRI, and the combination, respectively. On a lesional level, 85%, 83%, and 87% of the lesions were detected by PSMA-PET/CT, mpMRI, and the combination, respectively. For T-staging, mpMRI demonstrated better accuracy than PSMA-PET/CT in the detection of extraprostatic extension and seminal vesicle invasion. Combining PSMA-PET/CT with mpMRI did not yield a statistically significant improvement over using mpMRI alone. However, it is important to note that the study did not exclusively focus on clinically significant PCa. Herewith, the use of PSMA-PET offers only a slight improvement in the primary assessment of loco-regional findings. However, this could be of interest in the context of focal therapies and the use of image-guided intraprostatic radiation boosts [9].

Pilot studies are exploring the possibility of combining mpMRI and PSMA-PET to omit prostate biopsy for the confirmation of high-risk PCa in patients prior to radical prostatectomy (RP). When it comes to radiotherapy (RT), a biopsy remains essential for accurate dose determination. It is important to note that these findings are based on single-center studies with a limited number of patients included [30,31].

## 4. PSMA-PET in Primary Staging of PCa

The decision to conduct a staging investigation in patients with newly diagnosed PCa is based on risk group classifications, which consider a combination of PSA level, the biopsy-confirmed Gleason score (classified according to the International Society of Urological Pathology 2014 grade group system; ISUP), and the clinical examination (digital rectal examination) of the prostate. Patients are categorized into low-, intermediate- or high-risk groups for BCR, with minor variations depending on the classification system used [32,33,34]. PSMA-PET is becoming increasingly important for primary staging due to its higher sensitivity and specificity compared to conventional imaging techniques like (mp) MRI, CT, and BS. As a result, it is being gradually integrated into guidelines as an alternative to conventional imaging techniques, with differing indications, as shown in Table 1.

For low-risk PCa, staging is generally not recommended [32,33,35,36,37,38,39,40]. However, PSMA-PET/CT may improve active surveillance in low- to intermediate-risk PCa by detecting MRI-occult lesions and identifying patients at risk of pathological upstaging, as SUVmax correlates with higher percentages of aggressive Gleason patterns [41].

Around 5% to 10% of the newly diagnosed PCa patients are classified as cN1 cM0 by using CT and bone scintigraphy [32]. For intermediate-risk PCa patients, MRI during primary staging detects regional lymph node metastases in 4% and non-regional metastases in 0.8% of the cases. In high-risk patients, these figures rise to 11% for regional and 6% for non-regional lymph node metastases [42]. With [^11^C]Choline-PET, lymph node metastasis is identified in intermediate-risk patients with a region-based sensitivity and specificity of 8% and 99%, and a patient-based sensitivity and specificity of 19% and 90% [43]. Another study reports sensitivity and specificity of 50% and 76% in high-risk patients and of 71% and 93% in very high-risk patients for [^11^C]Choline-PET [44].

In contrast, PSMA-PET has been shown to be more efficient in detecting lymph node metastasis, as highlighted by several studies [9,45,46,47,48,49,50]. For patients with advanced PCa, [^68^Ga]Ga-PSMA-11-PET/CT can detect extraprostatic disease in 32% of the cases prior to primary treatment, even when conventional imaging with BS and pelvic CT/MRI reported negative results [45]. The proPSMA study, a paramount phase III randomized trial, compared conventional imaging (contrast CT of the abdomen/pelvis and SPECT/CT BS) with [^68^Ga]Ga-PSMA-11-PET/CT in 302 high-risk PCa patients. [^68^Ga]Ga-PSMA-11-PET/CT showed a 27% higher accuracy in detecting metastases, with improved sensitivity (85% vs. 38%) and specificity (98% vs. 91%) compared to conventional imaging. In patients with pelvic lymph node metastasis, [^68^Ga]Ga-PSMA-11-PET/CT demonstrated 32% greater accuracy (91% vs. 59%), and in those with distant metastasis, it showed 22% higher accuracy (95% vs. 74%). Moreover, [^68^Ga]Ga-PSMA-11-PET/CT produced fewer uncertain results (7% vs. 23%) and could also reduce the total exposure to radiation [50].

**Table 1 cancers-16-04263-t001:** Staging recommendations according to leading guidelines.

Guideline	Primary Staging		Biochemical Recurrence
	Intermediate Risk	High Risk	
EAU [32]	PSMA-PET if available, at least CI		After RP: PSMA-PET/CT in PSA > 0.2 ng/mL; [^18^F]Fluciclovine- or [^11^C]Choline-PET/CT if PSMA-PET not available and PSA ≥ 1 ng/mL After RT: PSMA-PET/CT if available in PSA +2.0 ng/mL over nadir or [^18^F]Fluciclovine- or [^11^C]Choline-PET/CT
AUA [33,40]	Favorable risk: no staging	PSMA-PET if CI is negative	Preferably PSMA-PET/CT, alternatively CI
	Unfavorable risk: staging may be considered (modality not stated)		
NCCN [36]	Favorable: no staging	PSMA-PET-CT/MRI or CI (CI not necessary before PSMA-PET)	PSMA-PET/CT/MRI or CI (CI not necessary before PET)
	Unfavorable: PSMA-PET/CT/MRI or CI (CI not necessary before PSMA-PET)		
S3 [35]	No statement possible due to lack of evidence, but stage with bone scintigraphy at PSA > 10	CI, PSMA-PET/CT possible	PSMA-PET/CT
ESMO [37]	CI or WB-MRI or PSMA-PET		PSMA-PET/CT
ASCO [38]	Favorable risk: no staging	PSMA-PET/CT or WB-MRI or [^18^F]-NaF-PET if CI is negative	WB-MRI, [^11^C]Choline-PET, [^18^F]Fluciclovine-PET, [^18^F]NaF-PET, or PSMA-PET if CI is negative
	Unfavorable risk: consider CI		
NICE [39]	No statement about PSMA-PET		Bone scan

CI: conventional imaging; WB-MRI: whole-body MRI.

A recent international collaborative review reported specificity rates for detecting lymph node metastasis between 70% and 100%, with most studies showing specificity > 90%. Sensitivity rates vary widely, from 10% to 100%, with well-designed prospective studies typically reporting a sensitivity of around 40%. Sensitivity is closely related to lymph node size: up to 93% for nodes > 10 mm and about 60% for nodes > 5 mm. This indicates that microscopic metastases are not well visualized with PSMA-PET imaging [9]. Due to the reduced sensitivity, which tends to increase with higher PSA levels, a negative PSMA-PET result should not automatically exclude the need for pelvic lymphadenectomy [15,46].

A meta-analysis by Tu et al. evaluated 904 patients across 11 studies who had undergone [^68^Ga]Ga-PSMA-11-PET/CT for untreated intermediate- or high-risk PCa prior to RP and pelvic lymphadenectomy. The comparison between preoperative imaging and histopathological findings revealed a pooled sensitivity and specificity of 63% and 93% in the per-patient analysis, and 70% and 99% in the per-node analysis, respectively [46]. A multicenter, prospective phase III imaging study evaluated patients with intermediate- and high-risk PCa prior to RP and pelvic lymph node dissection, finding a sensitivity of 40% and specificity of 95% for [^68^Ga]Ga-PSMA-11-PET [51]. In contrast, a meta-analysis by Hovels et al. found that primary staging with CT had a pooled sensitivity and specificity of 42% and 82%, while MRI achieved a pooled sensitivity and specificity of 39% and 82%, respectively [52].

When it comes to detecting distant metastasis (hormone-sensitive metastatic PCa), PSMA-PET/CT has also shown superiority over conventional imaging methods [9]. MRI can identify primary bone metastasis in 0.8% of the intermediate-risk patients and 10% of the high-risk patients [42]. BS offers sensitivity and specificity ranges of 61% to 89% and 61% to 96%, respectively. However, [^68^Ga]Ga-PSMA-11-PET/CT significantly outperforms MRI and BS, with sensitivity and specificity rates between 96% and 100% and 88% and 100%, respectively. Furthermore, PSMA-PET/CT reduces the number of ambiguous findings in bone metastasis cases and allows for a more precise determination of the number of bone metastases [53,54,55,56]. A multicenter retrospective study including 167 patients demonstrated a low positive predictive value of bone scans in patients undergoing initial staging, with 57% of the positive bone scans being identified as false positives in PSMA-PET. It is not specified whether [^18^F]- or [^68^Ga]Ga-PSMA-11-PET was used or if it was combined with CT or MRI [57]. A prospective study comparing methods for detecting bone metastasis found that whole-body MRI had sensitivity, specificity, and accuracy rates of 80%, 83%, and 82%, whereas NaF-PET/CT achieved 95%, 97%, and 96%, and [^68^Ga]Ga-PSMA-11-PET/CT reached 100% across all the metrics [58]. Another prospective study comparing [^18^F]-PSMA-1007-PET/CT with whole-body MRI, SPECT/CT, and CT highlighted the superior diagnostic accuracy of [^18^F]-PSMA-1007-PET/CT, but also emphasized the higher risk of false positive findings with [^18^F]-PSMA-1007-PET/CT, which are mostly due to unspecific bone uptake (UBU), as exemplarily shown in Figure 1 [59]. UBU corresponds to benign PSMA-positive lesions without morphological correlates on CT or MRI. They are defined as lesions with an SUVmax below 10, lacking both morphological features suggestive of metastatic disease and clear benign findings, such as inflammatory joint diseases, fractures, or fibrous dysplasia. The most common sites of UBUs, in descending order of frequency, are the ribs, pelvis, and spine [60]. [^18^F]-PSMA shows a higher benign bone uptake compared to [^68^Ga]Ga-PSMA-11 [17]. However, interpretation and reporting by experienced investigators do not appear to lead to increased detection of osseous metastases in [^18^F]-PSMA-1007-PET compared to [^68^Ga]Ga-PSMA-11-PET [61].

Visceral metastases predominantly impact the lungs and liver, but they are rarely identified during the initial diagnosis of PCa [62]. Consequently, there is limited information available on the initial detection of pulmonary and hepatic metastases using PSMA-PET imaging [63]. When it comes to lung metastases, 72.5% are PSMA-positive in [^68^Ga]Ga-PSMA-11-PET/CT, while 27.5% are PSMA-negative, complicating the distinction between metastases, benign lesions, and primary lung cancer [64,65]. Similarly, in the case of liver metastases, 77.7% of the metastatic lesions are PSMA-positive in [^68^Ga]Ga-PSMA-11-PET/CT, with 22.3% yielding false negative results [66]. Overall, the identification of lung and liver metastases is challenging due to varying PSMA expression, though it remains a secondary concern in primary diagnostics.

While PSMA-PET is effective for monitoring the progression of metastatic disease in patients undergoing androgen deprivation therapy (ADT), the number of positive findings correlates with rising PSA levels. Re-staging should ideally be conducted before the initiation of hormonal therapy given the risk of both false negative findings (as a result of therapy) as well as false positive results due to an imaging flare-up phenomenon when using GnRH receptor agonists. After establishing ADT, further imaging should be avoided within the first three months of treatment [14,16,67,68]. New approaches, such as the interpretation of SUV in [^68^Ga]Ga-PSMA-11-PET/CT as a prognostic biomarker due to its correlation with treatment response to hormonal therapy, show promise but require further investigation [69].

## 5. PSMA-PET After Treatment with Curative Intent

### 5.1. PSA-Persistence After RP

After RP, 5–20% of the patients experience PSA persistence (PSA ≥ 0.1 ng/mL 4–8 weeks post-surgery), which could be due to either residual tumor tissue, pre-existing metastases, or remaining benign prostatic tissue. Conventional imaging methods like MRI and bone scintigraphy have a low detection rate when PSA levels are below < 2 ng/mL. In contrast, PSMA-PET/CT can identify residual tumor activity at even lower PSA levels. [^68^Ga]Ga-PSMA-11-PET can detect tumors in 33% of the cases at PSA-levels < 0.2 ng/mL, in 46% of the cases at PSA-levels between 0.2 and 0.5 ng/mL, and in 97% of the cases with PSA > 2 ng/mL [32,70]. While mpMRI is superior to PSMA-PET/CT for detecting local lesions in the surgical bed, especially those near the bladder due to urinary tracer excretion, combining mpMRI and PSMA-PET imaging may improve residual lesion detection [71,72].

A multicenter study conducted a retrospective analysis of 191 high-risk patients with PSA persistence of ≥0.1 ng/mL after RP, finding that [^68^Ga]Ga-PSMA-11-PET/CT was able to detect PCa in 68% of these patients. Among them, only 33 had undergone a preoperative PSMA-PET/CT. In 45% of these cases, previously identified lesions were detected again, while in 21% of the patients, at least one new lesion was discovered postoperatively. Most of the detected lesions were located in the presacral/mesorectal and the area of the obturator lymph nodes [73]. According to the EAU guidelines, a PSMA-PET/CT should be performed when PSA persistence exceeds 0.2 ng/mL. If no metastases are found, local salvage RT with concurrent ADT is recommended. In cases where metastases are present, PSMA-PET/CT can help to guide localized salvage RT [74]. However, it is important to note that the absence of randomized controlled trials makes it difficult to provide clear treatment recommendations in the case of extrapelvic metastases [32].

### 5.2. Biochemical Recurrence

Following curative treatment with either RP or RT, 20–50% of the patients experience a BCR within 10 years [28]. BCR is defined as a PSA level rising above 0.2 ng/mL on two consecutive occasions after achieving undetectable levels after RP, or a PSA increase of more than 2 ng/mL above the nadir on two consecutive measurements following RT. BCR is detected rather late with BS. The likelihood of a positive bone scan is less than 5% when the PSA is <7 ng/mL [75,76]. Also, a CT scan is often ineffective in the early detection of metastases during BCR. Metastases are identified on CT scans in only 11–14% of the patients with BCR following RP [76]. According to the literature, a positive CT finding usually occurs when PSA levels exceed 27 ng/mL and the PSA doubling time (DT) is 1.8 ng/mL/month [77]. In contrast, [^11^C]Choline-PET and [^18^F]-Choline-PET provide superior diagnostic accuracy, with a sensitivity of 86-89% and specificity of 89-93% for detecting BCR [78,79]. Due to its higher efficacy, Choline-PET has been preferred over conventional imaging in earlier guidelines [80]. [^11^C]Choline-PET can detect bone metastases in up to 15% of the patients with BCR after RP who are negative on bone scintigraphy [81]. However, its efficiency is limited in identifying lymph node metastases, with pooled sensitivity and specificity rates of 62% and 92%, only [82]. In general, Choline-PET detection rates for BCR after RP range from 5 to 24% for PSA levels below 1 ng/mL and increase to 63-83% for PSA levels above 3 ng/mL, rising up to 67–100% for PSA levels above 5 ng/mL [78,83].

In contrast, an [^18^F]-PSMA-PET/CT scan demonstrated a 16% higher detection rate compared to [^18^F]-Choline-PET/CT (84% vs. 69%) in a prospective phase III study involving 186 PCa patients who had been treated with a curative intention and experienced BCR. Additionally, [^18^F]-PSMA-PET/CT showed superior detection at lower PSA levels. [^18^F]-PSMA-PET/CT achieved a detection rate of 52% at a PSA cutoff of ≤0.03 ng/mL, which was comparable to the detection rate of [^18^F]-Choline-PET/CT at a PSA level of ≤2.2 ng/mL. The [^18^F]-PSMA-PET/CT also exhibited a higher detection rate when considering PSA-DT: in cases of short PSA-DT of 0.3 months, [^18^F]-PSMA-PET/CT detected 83% of the cases compared to 76% with [^18^F]-Choline-PET/CT. With a longer PSA-DT (>12 months), [^18^F]-PSMA-PET/CT had a detection rate of 71% versus 48.4% in [^18^F]-Choline-PET/CT. Detection rates also correlated significantly with the initial TNM stage, with [^18^F]-PSMA-PET/CT outperforming [^18^F]-Choline-PET/CT in less aggressive T1 tumors, showing a detection rate of 73.1% compared to 42.3% for [^18^F]-Choline-PET/CT. For T3 tumors, the detection rate was 87.3% with [^18^F]-PSMA-PET/CT compared to 81.8% in [^18^F]-Choline-PET/CT. However, it is important to note that histological confirmation to rule out false positives was not performed in this study [84]. A comparable study demonstrated that [^68^Ga]Ga-PSMA-11-PET/CT achieved a detection rate of 50% at PSA levels < 0.5 ng/mL, whereas Choline-PET/CT detected 12.5% only in patients with T3 tumors [85]. In a publication including the histopathological confirmation of lymph node metastases, [^68^Ga]Ga-PSMA-11-PET/CT achieved a positive predictive value of 87% compared to 71% in [^18^F]-Choline-PET/CT [86].

In a direct comparison with CT, [^68^Ga]Ga-PSMA-11-PET/CT (in 36 patients) and [^18^F]-PSMA-PET/CT (in 23 patients) demonstrated a detection rate of 83–87% at a median PSA level of 1.96 ng/mL, whereas CT alone showed a detection rate of only 47–52% [87]. At a median PSA level of 0.32 ng/mL, [^18^F]-PSMA-PET/CT was able to identify metastasis in 46% of the patients, while CT detected metastasis in 16%, only [88].

To summarize, PSMA-PET has established itself as the imaging modality of choice in the diagnosis of BCR due to its diagnostic superiority over conventional imaging (see Table 1). While the ASCO guidelines suggest alternative options, they highlight the increasing significance of PSMA-PET, even though FDA approval was still pending at the time of their publication [38]. In the meantime, this approval has been granted [33]. On the other hand, the NICE guidelines released a statement in 2023 not to include PSMA-PET due to insufficient evidence. However, it noted that they will continue to monitor the emerging data [39].

## 6. PSMA-PET in Castration-Resistant PCa

Castration-resistant PCa (CRPC) is defined by biochemical or radiological progression despite suppressed testosterone levels. Biochemical progression is characterized as three consecutive rises in PSA levels, measured at least one week apart, resulting in two 50% increases from the nadir and a PSA level exceeding 2 ng/mL. Radiographic progression is defined as the appearance of two or more new bone lesions on a bone scan or the detection of a new soft tissue lesion according to the RECIST (Response Evaluation Criteria in Solid Tumours) criteria [89]. Although curative treatment is no longer an option at this stage, various palliative therapies are available. Treatment planning involves distinguishing between non-metastatic (nmCRPC) and metastatic CRPC (mCRPC), a differentiation defined by using BS and computed tomography [32,35].

A retrospective study involving 200 CRPC patients with PSA levels ≥ 2 ng/mL and high metastatic risk (PSA-DT ≤ 10 months under ADT or Gleason score ≥ 8) demonstrated the superiority of PSMA-PET compared to conventional imaging. Using conventional techniques (91% CT, 15% MRI, 11% bone scan/Choline-PET, and 3% other PET scans), these patients were initially classified as nmCRPC, defined as no pelvic lymphadenopathy ≥ 2 cm. Within three months, PSMA-PET imaging was conducted at a median PSA level of 5 ng/mL to detect the disease burden in 196 of 200 patients (98% detection in the PSA-DT ≤ 10 months subgroup and 100% in the Gleason score ≥ 8 subgroup). The findings revealed a local recurrence in 24% of the patients and pelvic-confined disease in 44%. Additionally, M1 disease was identified in 55% of the patients (58% in the PSA-DT ≤ 10 months subgroup and 49% in the Gleason score ≥ 8 subgroup). Among those with M1 disease, 39% had extrapelvic lymph node involvement, 24% bone metastases, and 6% visceral organ involvement. The extent of N/M disease detected by PSMA-PET was unifocal in 15%, oligometastatic (2–3 metastases) in 14%, and disseminated (≥4 lesions) in 46% of the patients. PSMA-PET imaging protocols were not standardized, as PSMA-PET/CT was used in 191 cases and PSMA-PET/MRI in 9 cases, employing [^68^Ga]Ga-PSMA-11 as the tracer in 195 cases and [^18^F]-DCFPyL in 5 cases [90].

Another study investigated 30 patients with CRPC initially undergoing therapy with curative intention. Two-third (n = 20) of the patients had PSA levels of ≥2 ng/mL, while 1/3 of the patients (n = 10) had a PSA value of ≤2 ng/mL including 7 patients with a PSA-DT of <12 months. Overall, 27 patients had a DT of <12 months. A recently performed [^18^F]-Choline-PET/CT scan, conducted within the past three months, had diagnosed these patients as nmCRPC. However, [^68^Ga]Ga-PSMA-11-PET/CT detected at least one focal lesion in overall 90% of the patients, which was detected in 100% of the patients with a PSA level of ≥2 ng/mL and 70% of the patients with a PSA level of ≤2 ng/mL (96% of the patients had a DT of <12 months), and in 33% of the patients with a PSA level of ≤2 ng/mL and a DT of >12 months. Three patients (10%) showed no detectable disease (with PSA levels of 0.3, 0.4, and 1.5 ng/mL, respectively), two patients (7%) had a local recurrence (with PSA levels of 1.6 and 3 ng/mL, respectively), six patients (20%) were classified as oligometastatic (PSA range: 0.4–21 ng/mL), and 19 patients (63%) were polymetastatic (PSA range: 0.9–90 ng/mL). Among the 10 patients with a PSA level of ≤2 ng/mL, three had no detectable disease, one had an isolated malignant focus in the prostate-specific area, three were oligometastatic, and three were polymetastatic [91].

A retrospective multicenter study examined 67 patients with CRPC. Conventional imaging was performed using computed tomography (CT) and bone scans (78% of the patients with a median PSA level of 122 ng/mL) or whole-body MRI and bone scans (22% of the patients with a median PSA level of 29 ng/mL). Within three months, a [^68^Ga]Ga-PSMA-11-PET scan was conducted (22% PET/MRI and 78% PET/CT, with a median PSA level of 53 ng/mL). Overall, conventional imaging showed positive findings in 87% of the patients, while [^68^Ga]Ga-PSMA-11-PET detected positive findings in 92%. The positivity rates for CT, BS, and whole-body MRI were 96%, 90%, and 47%, respectively. There was a discrepancy between [^68^Ga]Ga-PSMA-11-PET and conventional imaging in 30% of the patients. [^68^Ga]Ga-PSMA-11-PET led to an upgrade in 15% of the patients: four patients were upgraded from nodal or osseous to visceral metastasis, and in six patients the status shifted from non-metastatic to local recurrence (n = 2), nodal (n = 3), or osseous (n = 1) metastases. Conversely, [^68^Ga]Ga-PSMA-11-PET resulted in a downgrade in 15% of the patients: conventional imaging had diagnosed visceral disease in the lungs (n = 5), liver (n = 2), or adrenal glands (n = 1) in seven patients, but these findings were not confirmed by [^68^Ga]Ga-PSMA-11-PET. Moreover, three patients with assumed osseous metastasis in conventional imaging were not confirmed by [^68^Ga]Ga-PSMA-11-PET. Agreement between conventional imaging and [^68^Ga]Ga-PSMA-11-PET was particularly noted in patients with osseous metastasis (58% in conventional imaging, 60% in [^68^Ga]Ga-PSMA-11-PET). Additionally, almost all the patients with a PSA level above 15 ng/mL were classified as having multifocal metastases in both imaging modalities [92].

As PSA measurements alone are insufficient for mCRPC follow-up due to a possible lack of PSA expression in visceral metastases, the Prostate Cancer Working Group (PCWG) has developed evolving guidelines from PCWG1 to PCWG4 for assessing mCRPC [32,93]. These criteria incorporate RECIST for soft tissue lesions and introduce specific measures for bone metastases. While PCWG2 and PCWG3 emphasized bone scintigraphy and CT scans, the preliminary PCWG4 integrates PSMA-PET/CT imaging instead of bone scintigraphy for defining progression and response. For PSMA-PET/CT, new lesions indicate progression, while quantitative PET parameters (SUVmean, tumor volume) remain investigational. PSMA PET/CT demonstrates substantial agreement across response levels, enables earlier progression detection compared to PCWG3, and correlates well with overall survival [94,95,96]. Updated definitions are pending publication.

Despite the superiority of PSMA-PET in the context of CRPC, there are limitations in sensitivity and specificity, which can be attributed to the heterogeneity of CRPC, such as neuroendocrine dedifferentiation, resulting in differences in tracer uptake or histopathology between various metastases, along with a varied response to systemic therapies. In such cases, the use of dual-tracer imaging (i.e., using both PSMA-PET and FDG-PET) is an alternative option [40,97]. Imaging with somatostatin receptor targeting radioligands such as [^68^Ga]Ga-DOTATATE, or other ligands such as [^18^F]FDG or [^18^F]-Choline may be beneficial. The recently published multicenter prospective observational cohort study 3TMPO used a triple-tracer PET imaging approach to determine the prevalence of intrapatient intermetastatic heterogeneity and eligibility to radioligand therapy in 98 mCRPC patients with at least three active metastases on conventional imaging. The patients underwent [^18^F]FDG-PET/CT and [^68^Ga]Ga-PSMA-617-PET/CT scans, and an additional [^68^Ga]Ga-DOTATATE-PET/CT scan was performed if [^18^F]FDG-positive/[^68^Ga]Ga-PSMA-negative lesions were detected. A total of 82.7% of the patients showed intrapatient intermetastatic heterogeneity, which was associated with a significant decrease in median OS. A total of 53.1% of the patients were eligible for PSMA radioligand therapy, while none were eligible for DOTATATE radioligand therapy. The use of a multi-tracer approach might be helpful for precision medicine and personalized treatment of advanced prostate cancer [98].

Despite promising approaches, the actual impact of PSMA-PET imaging technique on patient prognosis in CRPC remains unclear [90,91,97]. Additionally, the use of [^68^Ga]Ga-PSMA-11-PET/CT is also discussed as a potential prognostic tool to assess response and predict overall survival (OS) of therapy with androgen receptor signaling inhibitors (ARPI) in mCRPC [67].

## 7. Radioligand Therapy

There are various so-called theranostic pairs of PSMA ligands enabling imaging with, e.g., positron-emitting nuclides on the one hand and therapy with the (ideally pharmacologically identical) ligand equipped with a β- or α-particle-emitting nuclide such as [^177^Lu]Lu-PSMA-I&T. For patients with mCRPC progression following treatment with ADT, ARPI, and taxane chemotherapy, radioligand therapy using the low-energy β-particle-emitting radiopharmaceutical [^177^Lu]Lu-PSMA-617 has been shown to be life-prolonging compared to the previous standard-of-care. This therapy is feasible when PSMA-PET/CT detects one or more lesions with high PSMA expression (greater SUV than in the liver) [32,35,40]. An example of a patient with mCRPC and osseous metastases before and after 6 cycles of [^177^Lu]Lu-PSMA therapy is exemplarily shown in Figure 2. For effective therapy planning, it is recommended to adhere to specific criteria when using [^68^Ga]Ga-PSMA-11 and [^18^F]-PSMA [20,99].

In the international open-label phase III VISION study that ultimately led to the FDA approval of [^177^Lu]Lu-PSMA-617, patients pretreated with ARPI and taxane chemotherapy with at least one PSMA-positive metastasis and no PSMA-negative metastases detected by [^68^Ga]Ga-PSMA-11-PET/CT were included. A total of 280 patients received standard therapy (excluding chemotherapy, immunotherapy, Radium-223, or investigational drugs), while 581 patients received [^177^Lu]Lu-PSMA-617 in addition to standard therapy. Lutetium therapy achieved progression-free survival (PFS) of 8.7 months compared to 3.4 months with standard therapy alone. Median OS was also improved in the Lutetium group, with 15.3 months compared to 11.3 months in the standard therapy group. PSA reduction was assessed in 385 patients in the Lutetium therapy group compared to 196 patients in the standard therapy group, showing superior results in the Lutetium group (PSA reduction of ≥50% in 46% vs. 7% and PSA reduction of ≥80% in 33% vs. 2%). Among the 248 patients with measurable target lesions according to the RECIST criteria, a complete radiologic response was observed in 9.2% of the Lutetium group, while no complete responses were seen in the standard therapy group. A partial radiologic response was noted in 41.8% of the Lutetium group compared to 3% in the standard therapy group. A higher SUV correlated with a better response to Lutetium therapy. Grade 3 or higher adverse events occurred more frequently in the Lutetium group than in the standard therapy group (52.7% vs. 38.0%), although these did not negatively impact the patients’ quality of life. The therapy was discontinued in 11.9% of the patients in the Lutetium group due to side effects [100].

The multicenter, open-label, randomized phase II TheraP study compared patients with progression after docetaxel chemotherapy, 91% of whom had also been treated with ARPI. A total of 99 patients received [^177^Lu]Lu-PSMA-617 therapy following imaging with [^68^Ga]Ga-PSMA-11-PET/CT or [¹⁸F]FDG-PET/CT, while 101 patients received Cabazitaxel chemotherapy. Among the 78 patients with measurable lesions according to RECIST criteria, a response was observed in 49% of the Lutetium group compared to 24% in the Cabazitaxel group. As in other studies, a higher SUV correlated with an increased response rate in the Lutetium therapy group. A PSA reduction of 50% was achieved in 66% of the Lutetium group versus 37% of the Cabazitaxel group. The rate of Grade 3 or higher adverse events was 33% in the Lutetium group compared to 53% in the Cabazitaxel group. Patients in the Lutetium group also reported an improved quality of life. However, there was no significant difference in OS between the two groups, with a restricted mean survival time of 19.1 months in the Lutetium group and 19.6 months in the Cabazitaxel group [101,102]. A systematic review analyzing 40 retrospective and prospective studies highlighted that [^177^Lu]Lu-PSMA-617 therapy is associated with an improvement and sustained maintenance of quality of life. The review also found that a higher mean SUV correlates with a better PSA response to Lutetium therapy: an SUVmean ≥ 10 is associated with a 91% PSA response rate, whereas an SUVmean < 10 corresponds to a PSA response rate of only 52% [103].

Lutetium therapy may also be a viable treatment option for patients who have not yet undergone taxane therapy, although no phase III studies are available to date. Preliminary data of the SPLASH phase III trial, evaluating the efficacy and safety of [^177^Lu]Lu-PSMA-I&T in patients with mCRPC progressing after ARPI treatment, were recently presented. A total of 412 patients were randomized 2:1 to receive either [^177^Lu]Lu-PSMA-I&T (experimental group) or an alternative ARPI (enzalutamide or abiraterone; control group). Crossover to [^177^Lu]Lu-PSMA-I&T was permitted for patients in the control group upon radiographic progression, with 85% of those in the control group transitioning to [^177^Lu]Lu-PSMA-I&T. At a median follow-up of approximately 12 months, the trial met its primary endpoint by showing a radiographic progression-free survival advantage for [^177^Lu]Lu-PSMA-I&T (median of 9.5 months versus 6 months). A complete response was achieved by 9.3% of the patients in the experimental arm, compared to none in the control arm. Median response durations were 9.4 and 7.3 months, respectively. [^177^Lu]Lu-PSMA-I&T reduced the risk of radiographic progression or death by 29% compared to the control group while also showing a favorable safety profile (Grade ≥3 events in 10% of the patients in the experimental and 12% in the control group) [104]. In a randomized, open-label phase II study, 35 chemotherapy-naïve patients with mCRPC and highly PSMA-expressing lesions identified on [^68^Ga]Ga-PSMA-11-PET/CT were evaluated. Fifteen patients were assigned to the [^177^Lu]Lu-PSMA-617 arm, and 20 patients to the docetaxel arm. A PSA reduction of ≥50% was achieved in 60% of the Lutetium group compared to 40% in the docetaxel group. Fewer Grade 3 or higher adverse events occurred in the Lutetium group (30% vs. 50%). The median OS was 15 months in both groups. Additionally, more patients were able to switch from Lutetium to docetaxel therapy than vice versa [105]. A systematic review and meta-analysis comparing [^177^Lu]Lu-PSMA-ligand and taxane chemotherapy in taxane-naïve mCRPC patients included 24 studies for [^177^Lu]Lu-PSMA-ligand and 17 for taxane treatment. The median proportion of patients achieving at least a 50% decline in serum PSA from baseline was comparable between the groups (48.4% vs. 54.4%). However, median progression-free survival (5.5 vs. 7.4 months) and overall survival (13.8 vs. 20.6 months) were shorter in the [^177^Lu]Lu-PSMA-ligand group. With a shorter median follow-up (12.9 vs. 21.3 months), [^177^Lu]Lu-PSMA-ligand radioligand therapy shows promise for patients unable or unwilling to undergo chemotherapy [106].

The UpFrontPSMA trial, a multicentre, open-label, randomized phase II study, evaluated the combination of [^177^Lu]Lu-PSMA-617 and docetaxel in patients with de novo high-volume metastatic hormone-sensitive prostate adenocarcinoma who had received less than four weeks of androgen deprivation therapy (ADT). A total of 130 patients were enrolled and randomly assigned: 63 (48%) to receive [^177^Lu]Lu-PSMA-617 followed by docetaxel, and 67 (52%) to receive docetaxel alone, both groups continuing on ADT. At 48 weeks, 41% of the patients in the [^177^Lu]Lu-PSMA-617 plus docetaxel group achieved undetectable PSA levels, compared to 16% in the docetaxel-alone group. Serious adverse events occurred in 25% of the patients in both groups, with none definitively linked to [^177^Lu]Lu-PSMA-617 [107]. Another study investigating the potential use of Lutetium therapy in patients with hormone-sensitive metastatic PCa is the ongoing phase III PSMAddition study, assessing the use of [^177^Lu]Lu-PSMA-617 plus standard of care (ADT and ARPI) versus standard of care alone [108].

The earliest application of Lutetium within the treatment pathway of PCa was investigated in the phase I/II LuTectomy trial with the administration of [^177^Lu]Lu-PSMA-617 in a neoadjuvant setting for high-risk localized PCa 6 weeks prior to RP. Twenty patients were enrolled, with a median PSA of 18 ng/mL, 90% having ISUP grade group ≥3, and 30% presenting with N1 disease. [^177^Lu]Lu-PSMA-617 was shown to deliver high targeted radiation doses to sites of tumor with high PSMA expression. In 45% of the patients, a decline in PSA > 50% was observed with only mild treatment-related adverse events and no Grade 3/4 toxicities or severe surgical complications (Clavien–Dindo Grades 3–5). The feasibility trial is strongly limited by a small sample size, but the findings support further exploration of this therapy to evaluate its potential long-term oncological benefits [109].

In addition to Lutetium therapy, there is growing interest in α-emitting isotopes such as [^225^Ac]Ac or [^212^Pb]Pb and Auger-electron emitters such as [^161^Tb]Tb, which, due to higher and shorter range energy transfer, is expected to have greater deleterious effects on cancer cells with fewer unwanted effects on the surrounding tissue while potentially limiting the development of therapy resistance. A systematic review with the meta-analysis of nine studies involving 263 patients assessed the therapeutic effects of [^225^Ac]Ac-PSMA radioligand therapy in patients with mCRPC who had previously been treated with chemotherapy, [^177^Lu]Lu-PSMA-617, and/or [223Ra]Ra. The pooled proportion of patients with any PSA reduction was 84%, and those with a PSA reduction > 50% was 61%. The estimated median PFS and median OS after [^225^Ac]Ac-PSMA radioligand therapy were 9.15 months and 11.77 months, respectively. The authors state superior results for PSA-reduction compared to a previous meta-analysis for [^177^Lu]Lu-PSMA-617 (46% PSA-reduction) with similar PFS (11 months) and OS (14 months) [110]. The pooled incidence of side effects of [^225^Ac]Ac-PSMA radioligand therapy included 63% for Grade 1 or 2 xerostomia, 14% for Grade 3 or 4 anemia, 7% for Grade 3 or 4 thrombocytopenia, and 4% for Grade 3 or 4 leukopenia, which were reported as comparable to the side effect profile of [^177^Lu]Lu-PSMA-617 therapy. These findings suggest that [^225^Ac]Ac-PSMA-RLT may serve as an effective additional treatment option for patients with mCRPC. However, confirmation through prospective, multicenter, and randomized controlled studies is still needed [111].

## 8. Discussion

In terms of PCa diagnostics, PSMA-PET imaging offers significantly enhanced diagnostic accuracy compared to conventional imaging methods. Consequently, PSMA-PET is increasingly recommended in the leading international guidelines for the primary staging of intermediate- and high-risk PCa as well as for the diagnosis of BCR (Table 1). PSMA-PET also outperforms the diagnostic accuracy of conventional imaging and Choline-PET in CRPC, although it has limitations in the cases of neuroendocrine-differentiated PCa.

Regarding primary staging, there is considerable enthusiasm in the literature about the enhanced capabilities of PSMA-PET imaging. However, despite—or rather precisely because of—its advantages, PSMA imaging must be viewed with caution. Most studies in PCa are based on conventional imaging, as, for example, the phase III studies CHAARTED (Chemohormonal Therapy vs. Androgen Ablation Randomized Trial for Extensive Disease in Prostate Cancer) and LATITUDE (Latitude Trial: A Trial of Intensive Tumor Suppression with Abiraterone Acetate) [112,113]. The treatment-defining high-volume and high-risk categories established in these studies rely on conventional imaging. Despite attempts to integrate these criteria with PSMA-PET imaging, the comparison remains controversial and it is still a matter of debate on how to establish an appropriate threshold in PSMA-PET [114,115]. Additionally, while PSMA-PET detects more lesions with fewer false positives, careful attention is needed as false positives still occur [50,116]. The phenomenon of UBU remains a diagnostic challenge with PSMA PET. It can be difficult to differentiate between a bone metastasis and a UBU. In such cases, it is important not to deprive the patient of a potentially curative therapeutic option due to UBU in staging and restaging for BCR [60]. A detailed understanding of the significance of findings in PSMA-PET, reported by specialists with the necessary training and expertise, is, therefore, warranted to avoid over- or undertreatment.

Until now, there are relatively few studies that have prospectively examined the long-term impact of PSMA-PET imaging regarding treatment management and oncological outcomes [32,35,37,40]. The recently published PROMISE-PET study, a multicenter retrospective analysis including 2414 patients with PCa, aimed to assess the prognostic value of PSMA-PET using PROMISE criteria, demonstrating its potential as a novel biomarker for overall survival in prostate cancer across all stages. The study developed both visual and quantitative nomograms and seeks to implement novel PET-based risk tools to guide clinical management and study designs. In head-to-head comparisons with the established clinical risk scores and nomograms (e.g., the EAU biochemical recurrence subgroup), the visual PPP nomogram demonstrated the superior prediction of overall survival, though including a higher proportion of patients with advanced disease, potentially introducing selection bias. Other limitations include the lack of prospective validation, international cohorts, and cancer-specific survival data. Future efforts within the ongoing multicenter PROMISE-PET registry (NCT06320223) will collect larger datasets to optimize nomograms and assess long-term outcomes, particularly for initial staging, biochemical recurrence, and non-mCRPC cases, further evaluating PSMA-PET as a promising tool for determining oncological outcomes [117].

Regarding therapeutic use, [^177^Lu]Lu-PSMA-617 therapy can provide a significant extension of the OS in certain patient groups with mCRPC as an addition to standard therapy. It also offers improved quality of life as an alternative to second-line taxane chemotherapy [101,102]. Promising results have been reported in a phase II study and the preliminary results of a phase III study on Lutetium therapy in taxane-naïve mCRPC patients [104,105]. Additionally, phase II studies have also been conducted in hormone-sensitive metastatic settings, and the earlier application of Lutetium therapy in this population is currently being investigated in an ongoing phase III study [107,108].

In a systematic review with a meta-analysis by Jeet et al., 54 studies were summarized, revealing that PSMA-PET/CT ([^68^Ga]Ga-PSMA-11 in 81% and [^18^F]-PSMA in 13% of the studies) led to changes in clinical management in 28% of the cases with primary PCa and in 54% of the cases with recurrent disease [15]. The largest phase III study to date on PSMA imaging in primary high-risk PCa demonstrated that [^68^Ga]Ga-PSMA-11-PET/CT resulted in relevant changes to treatment management in 27% of the patients. However, the study emphasized the lack of evidence regarding improved outcomes [50]. The prospective phase III PSMA-dRT trial aimed to assess the impact of PSMA-PET/CT (both [^18^F]- and [^68^Ga]Ga-PSMA-11) on treatment planning before RT and to analyze outcomes over five years. During the study, FDA approval and insurance coverage for PSMA-PET/CT were granted, making the control arm without PSMA-PET/CT increasingly untenable for many patients and physicians. Consequently, the study was terminated after enrolling only 54 patients. Of these, 17% were upgraded compared to conventional imaging and received MDT. No usable data on outcomes are available [118].

In the metastatic setting, the enhanced detection of metastases through PSMA-PET/CT is highlighted as a significant advantage in planning MDT [119]. However, the results from prospective studies evaluating the oncological outcomes are still lacking [32,35,37,40]. For instance, there is evidence suggesting that PSMA-PET/CT may improve the outcome in patients with BCR after RP by preventing salvage RT and administering MDT detected by PSMA-PET/CT. A retrospective comparison of two cohorts, each with 108 patients, demonstrated that staging with PSMA-PET/CT without evidence of metastatic disease at BCR led to a 62% reduction in the relative risk of biochemical progression one year after salvage RT compared to “blind” salvage RT without conventional staging. However, it is important to note that the radiation dose in the PSMA cohort was significantly higher at 70 Gy, compared to 66 Gy in the “blind” cohort [120].

In CRPC, improved imaging through PSMA-PET/CT has led to a significant reduction in the number of M0-CRPC patients, making the presence of nmCRPC increasingly rare while leading to the Will Rogers phenomenon. This phenomenon, which can be broadly applied to PSMA-PET diagnostics, arises from stage migration due to enhanced detection. Survival rates improve in the healthier patient group because the least healthy patients are reclassified to a more advanced stage. Similarly, the less healthy group shows better survival rates as patients are identified and treated earlier in their disease progression [121]. While PSMA-PET/CT and PSMA-PET/MRI allow for better diagnosis of oligometastatic CRPC and facilitate PET-guided stereotactic treatments, enabling the switch between RT and systemic treatments as needed, most studies to date are retrospective. Consequently, no definitive conclusions can yet be drawn about the improvement of oncological outcomes in CRPC based on PSMA-PET [90,97].

Until relevant data from prospective and multicenter clinical studies are available, decision making based on PSMA-PET imaging should still be approached with caution, as most large prospective treatment studies have been conducted based on conventional imaging. Significant changes to treatment regimens based on the improved diagnostics of PSMA-PET must be made in an interdisciplinary setting and tailored to the individual patient’s needs.

## 9. Conclusions

In conclusion, PSMA-PET imaging has shown substantial advantages in the diagnostic workup of localized and advanced PCa. It stands out by allowing more precise staging and evaluation of biochemical recurrence and metastatic disease, contributing to more targeted therapeutic decisions. This review highlights the benefits of PSMA-PET imaging, its implementation across various guidelines, and its role in enhancing patient stratification for treatments like MDT and PSMA radioligand therapy, which represents a breakthrough in the treatment of mCRPC. However, precisely because of the diagnostic advantages, there remains a need for cautious implementation in clinical settings. Many existing treatment pathways have been developed around conventional imaging methods and cannot be directly translated to PSMA-based approaches without careful consideration. The enthusiasm surrounding PSMA-PET and its broad implementation makes it challenging to conduct prospective comparative studies with conventional imaging. Until robust data are available on long-term oncological outcomes, treatment decisions based on PSMA-PET findings should be made thoughtfully, within a multidisciplinary framework.

## Figures and Tables

**Figure 1 cancers-16-04263-f001:**
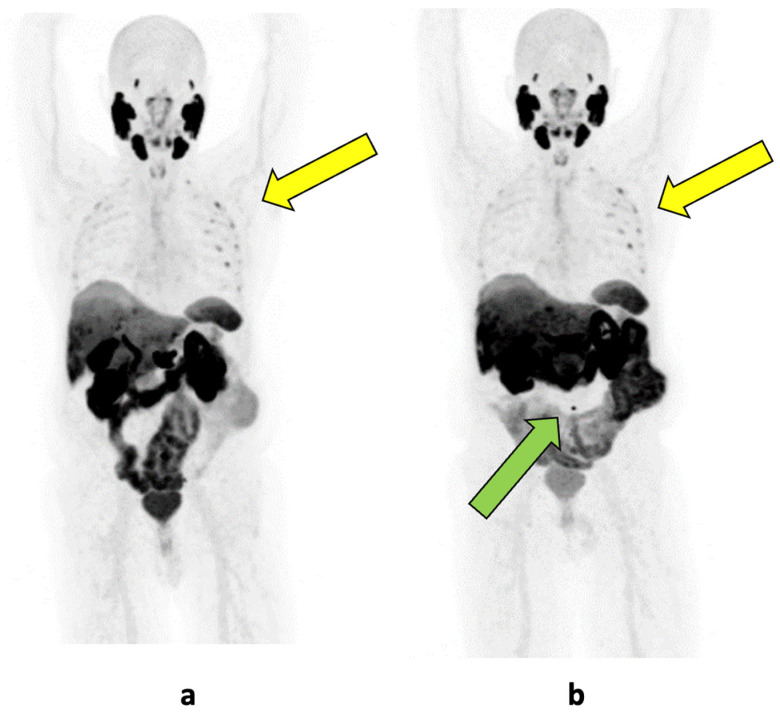
Maximum intensity projection images of [^18^F]-PSMA PET/CT scans of a patient with biochemical recurrence. (**a**) Imaging at a PSA level of 0.04 ng/mL showing unspecific bone uptake in the ribs (yellow arrow); (**b**) follow-up imaging after a PSA increase to 0.08 ng/mL, diagnosing a lymph node metastasis (green arrow) as an explanation for the rising PSA levels, with persistent unspecific bone uptake (yellow arrow).

**Figure 2 cancers-16-04263-f002:**
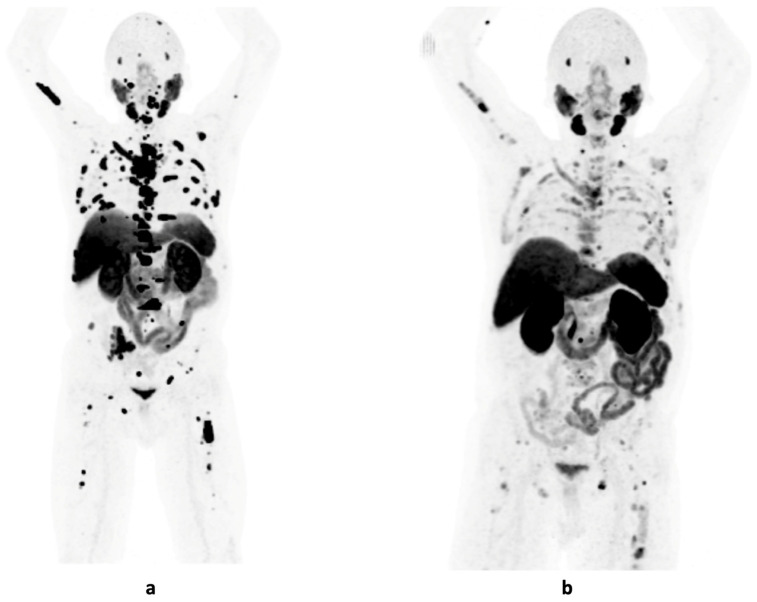
Maximum intensity projection images of the [^18^F]-PSMA PET/CT scans of a patient with mCRPC and osseous metastases at an initial PSA level of 128 ng/mL (**a**); follow-up imaging after 6 cycles of [^177^Lu]Lu-PSMA therapy with a PSA reduction to 3.3 ng/mL (**b**).

## Data Availability

No new data were generated or collected in this study.

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
