# Peer review of "Current Clinical Applications of PSMA-PET for Prostate Cancer Diagnosis, Staging, and Treatment"

_cancers, 2024, doi:10.3390/cancers16244263_

Round 1
Reviewer 1 Report
Comments and Suggestions for Authors
This review was well written and highlighted the benefits of PSMA-PET in patients with prostate cancer. The authors reviewed extensive updated research papers and made many fair comparison and conclusions.
1. Under “4. PSMA-PET in primary staging of PCa”. We all believed that PSMA PET is better in detecting LN mets or bone mets than MRI or CT. But staging also includes evaluation of the prostate capsule and seminal vesicle involvement. The authors need to talk about mpMRI’s role for this issue.
2. Under “5.1. PSA-persistence after RP”. Since PSMA accumulates strong activity in the bladder, it is not easy to detect small local recurrence following RP. Therefore, in US, we routinely use mp-MRI to detect local recurrence, particularly relying on dynamic contrast imaging. Can authors discuss this?
3. Under “5.1. PSA-persistence after RP”, second paragraph, line 284-295. Authors discussed detection of residual prostate cancer after RP, but mentioned detection residual PCa in the presacral space or pelvic LNs. In US, when we talked about residual disease, we meant residual tumor in the surgical bed. Please clarify.
4. SUV is very important measurement in PSMA PET, higher, ? more aggressive. The authors did not mention meaning of high SUV in the positive lesions.
5. There are false positive or negative in PSMA PET. Can authors talk a little bit? Thanks.
Author Response
We would like to sincerely thank the reviewers for taking the time to read our manuscript and for their valuable and insightful comments. We greatly appreciate the effort and time dedicated to providing this feedback, and we are pleased to have the opportunity to improve our paper based on the reviewers’ suggestions. The implementation of their comments is addressed and explained under each bullet point in response to their respective questions below.
Reviewer 1:
This review was well written and highlighted the benefits of PSMA-PET in patients with prostate cancer. The authors reviewed extensive updated research papers and made many fair comparison and conclusions.
- Under “4. PSMA-PET in primary staging of PCa”. We all believed that PSMA PET is better in detecting LN mets or bone mets than MRI or CT. But staging also includes evaluation of the prostate capsule and seminal vesicle involvement. The authors need to talk about mpMRI’s role for this issue.
- We thank Reviewer #1 for his important comment. The role of mpMRI and a comparison with PSMA-PET is discussed under “3. Assessment of loco-regional findings and prostate biopsy planning”. Please see pages 3-4, lines 122-125 and 136-157.
- Under “5.1. PSA-persistence after RP”. Since PSMA accumulates strong activity in the bladder, it is not easy to detect small local recurrence following RP. Therefore, in US, we routinely use mp-MRI to detect local recurrence, particularly relying on dynamic contrast imaging. Can authors discuss this?
- Please find our combined answer to comments number 2. and 3. under response 3.
- Under “5.1. PSA-persistence after RP”, second paragraph, line 284-295. Authors discussed detection of residual prostate cancer after RP, but mentioned detection residual PCa in the presacral space or pelvic LNs. In US, when we talked about residual disease, we meant residual tumor in the surgical bed. Please clarify.
- Thank you for your comments regarding the terminology of “local recurrence” and “residual disease”. Under “5.1. PSA-persistence after RP” (page 7, lines 295-297), it is stated that PSA persistence may result from residual tumor tissue, pre-existing metastases, or remaining benign prostatic tissue. The cited study examined patients with PSA persistence and identified these causes. The term “residual” (page 7, line 307) was though misleading and removed. Another paragraph was added focusing on residual disease in the surgical bed, also addressing your second comment (page 7, lines 301-304).
- SUV is very important measurement in PSMA PET, higher, ? more aggressive. The authors did not mention meaning of high SUV in the positive lesions.
- We share the relevant thoughts of Reviewer #1. A statement about the relation of high SUV was given under “3. Assessment of loco-regional findings and prostate biopsy planning”. Herewith, the sentence was moved to another position (under “2. Imaging modalities and PSMA targeting in PCa”, page 3, lines 117-119) and further extended to substantiate it. Additionally, the prognostic value of SUV is discussed under “7. Radioligand therapy” (page 11, lines 513, 525-526, 534-537).
- There are false positive or negative in PSMA PET. Can authors talk a little bit? Thanks.
- This is an important topic and an issue that concerns many diagnostic tests. We believe that false positive and false negative results are already discussed thorougly, e.g. on page 6, lines 250-262; page 6, lines 275-280 and page 14, lines 641-648.
Reviewer 2 Report
Comments and Suggestions for Authors
Well written and comprehensive review.
I suggest to the authors to expand the section on the new role of PSMA guided biopsy (PSMA PET-targeted Biopsy for Prostate Cancer Diagnosis: Initial Experience From a Multicenter Cohort. Urology. 2024 Oct 18:S0090-4295(24)00909-9. doi: 10.1016/j.urology.2024.10.026. Epub ahead of print. PMID: 39426743.)
Furthermore to expand the discussion on the role of stadiative PSMA PET in low/intermediate risk Pca
Finally the role of PSMA in upstaging the disease respect the standard imagination and this impact on BCR and survival.
Author Response
We would like to sincerely thank the reviewers for taking the time to read our manuscript and for their valuable and insightful comments. We greatly appreciate the effort and time dedicated to providing this feedback, and we are pleased to have the opportunity to improve our paper based on the reviewers’ suggestions. The implementation of their comments is addressed and explained under each bullet point in response to their respective questions below.
Reviewer 2:
Well written and comprehensive review.
I suggest to the authors to expand the section on the new role of PSMA guided biopsy (PSMA PET-targeted Biopsy for Prostate Cancer Diagnosis: Initial Experience From a Multicenter Cohort. Urology. 2024 Oct 18:S0090-4295(24)00909-9. doi: 10.1016/j.urology.2024.10.026. Epub ahead of print. PMID: 39426743.)
- We thank Reviewer #2 to mention this relevant study. It was added under “3. Assessment of loco-regional findings and prostate biopsy planning” (page 3, lines 125-234) and now enhances the evidence to our cited references.
Furthermore to expand the discussion on the role of stadiative PSMA PET in low/intermediate risk Pca
- Thank you, this is an important comment and will certainly require further research in the future. As PSMA-PET staging is not advised in low-risk disease, data is scarce. Nevertheless, an outlook for using PSMA-PET in active surveillance for low- and favorable intermediate-risk PCa was added on page 4, lines 176-179.
Finally the role of PSMA in upstaging the disease respect the standard imagination and this impact on BCR and survival.
- This is an important topic and needs to be discussed thoroughly. We believe, that this matter and especially the potential long-term impact of PSMA-PET imaging is well elaborated on page 10 lines 424-444; page 14, lines 631-641; pages 14-15, lines 650-666; page 15, lines 677-690; pages 15-16, lines 704-723 and page 16, lines 731-739.
Reviewer 3 Report
Comments and Suggestions for Authors
This is a non systematic review analysing the available evidence on PSMA PET/CT and radiopharmaceutical therapy with PSMA-labelled radiopharmaceuticals.
Comments:
1. Terminology:
1.1. The terminology used regarding PET is incomplete. Every time PET is mentioned in the text it should be indicated if PET/CT or PET/MR are considered. Nowadays, 99% of the systems are multimodality PET/CT or PET/MR systems, with stand-alone PET systems being exceptional. The acronym PET should be defined the first time it appears.
1.2 Regarding PSMA, the first time the acronym appears in the text it should be defined. Also, when referring to a generic PSMA-labelled radiopharmaceutical I suggest using PSMA-ligand or PSMA-X explaining it the first time it appears. When the radiopharmaceutical used is known it should be indicated, as has already been done in great part of the manuscript.
1.3 Instead of radioligand therapy I recommend using radiopharmaceutical therapy.
2. Evidence:
2.1 Page 12, lines 520: Taxane-naïve mCRPC patients: There are some studies mentioned here. I recommend commenting a very recent meta-analysis in this particular clinical setting: Almeida LS, García Megías I, Etchebehere ECSC, Calapaquí Terán AK, Herrmann K, Giammarile F, Treglia G, Delgado Bolton RC. Assessment of the therapeutic efficacy of [177Lu]Lu-PSMA-X compared to taxane chemotherapy in taxane-chemo-naïve patients with metastatic castration-resistant prostate cancer: A systematic review and meta-analysis. Eur J Nucl Med Mol Imaging. 2024 Oct 25. doi: 10.1007/s00259-024-06932-2. Epub ahead of print. PMID: 39453485.
3. Editorial aspects:
Page 3, line 95: Correct the following "2'104" to "2,104"
Author Response
We would like to sincerely thank the reviewers for taking the time to read our manuscript and for their valuable and insightful comments. We greatly appreciate the effort and time dedicated to providing this feedback, and we are pleased to have the opportunity to improve our paper based on the reviewers’ suggestions. The implementation of their comments is addressed and explained under each bullet point in response to their respective questions below.
Reviewer 3:
This is a non systematic review analysing the available evidence on PSMA PET/CT and radiopharmaceutical therapy with PSMA-labelled radiopharmaceuticals.
Comments:
- Terminology:
1.1. The terminology used regarding PET is incomplete. Every time PET is mentioned in the text it should be indicated if PET/CT or PET/MR are considered. Nowadays, 99% of the systems are multimodality PET/CT or PET/MR systems, with stand-alone PET systems being exceptional. The acronym PET should be defined the first time it appears.
- Thank you very much for your important comment. A consistent and comprehensible terminology is indispensable. Please find our combined answer to comments number 1.1 and 1.2 under response 1.2.
1.2 Regarding PSMA, the first time the acronym appears in the text it should be defined. Also, when referring to a generic PSMA-labelled radiopharmaceutical I suggest using PSMA-ligand or PSMA-X explaining it the first time it appears. When the radiopharmaceutical used is known it should be indicated, as has already been done in great part of the manuscript.
- All acronyms are now defined at first appearance. Whenever a study specified whether PSMA-PET/CT or PSMA-PET/MR was used, this distinction was clarified. In cases where such a statement was missing, it was also addressed, as seen, for example on page 6, lines 245-247. Due to the heterogeneity of the literature, a more general approach was required in the introduction and discussion, so PSMA-PET imaging was used in a summarized manner, as it is common in the literature. We chose not to adopt PSMA-X as it has not been widely utilized in the literature.
1.3 Instead of radioligand therapy I recommend using radiopharmaceutical therapy.
- Thank you for your valuable comment. Both terms, "radioligand therapy" and "radiopharmaceutical therapy," are commonly used in the literature. However, we have decided to adhere to the term "radioligand therapy" in alignment with the EAU guidelines and the ASCO guidelines, as well as most of the cited references.
- Evidence:
2.1 Page 12, lines 520: Taxane-naïve mCRPC patients: There are some studies mentioned here. I recommend commenting a very recent meta-analysis in this particular clinical setting: Almeida LS, García Megías I, Etchebehere ECSC, Calapaquí Terán AK, Herrmann K, Giammarile F, Treglia G, Delgado Bolton RC. Assessment of the therapeutic efficacy of [177Lu]Lu-PSMA-X compared to taxane chemotherapy in taxane-chemo-naïve patients with metastatic castration-resistant prostate cancer: A systematic review and meta-analysis. Eur J Nucl Med Mol Imaging. 2024 Oct 25. doi: 10.1007/s00259-024-06932-2. Epub ahead of print. PMID: 39453485.
- Thank you very much for mentioning this relevant study, which represents an important addition to our paper. Discussion and reference of the meta-analysis was added on page 13, lines 568-576.
- Editorial aspects:
Page 3, line 95: Correct the following "2'104" to "2,104"
- Thank you for noticing. This was corrected accordingly.